# Characteristics of non-drinking and drinking adults in Taiwan and their implications in alcohol epidemiological studies

Tsung Yu[1], Tzu-Jung Wong[2], Hsing-Yu Yang[3]*

1 Department of Public Health, College of Medicine, National Cheng Kung University, Tainan, Taiwan,
2 Department of Healthcare Information and Management, School of Health and Medical Engineering, Ming Chuan University, Taipei, Taiwan, 3 Department of Nursing, MacKay Medical College, New Taipei, Taiwan

* hyyang@mmc.edu.tw

## Abstract

### Background

Confounding is a major threat to causal inferences regarding alcohol and health. One proposed causal inference approach was to compare the associations obtained from cohorts in different countries, where we know the confounding structures are different. To explore the confounding structure related to alcohol and health in Taiwan, we compared differences in the characteristics of drinkers and non-drinkers, using data from the Nutrition and Health Survey in Taiwan (2013-2016).

### Methods

The cross-sectional survey data were collected nationwide and were representative of residents in Taiwan, which included 2,846 men and 2,833 women ages 19 years and older. We used information regarding alcohol drinking and covariates of sociodemographic and health behavior variables. We performed sex-stratified multinomial logistic regression to assess the associations between each independent variable and alcohol drinking (non-drinkers, occasional drinkers, and moderate drinkers).

### Results

Among men, 31% were non-drinkers, 7% former drinkers, 43% occasional drinkers, 13% moderate drinkers, and 5% heavy drinkers. Among women, 62% were non-drinkers, 1% former drinkers, 32% occasional drinkers, 4% moderate drinkers, and 1% heavy drinkers. In both men and women, occasional drinkers had higher educational levels and fewer family financial problems, but had more smoking and betel nut chewing behaviors than non-drinkers, after we adjusted for age. Moderate drinkers had even more smoking and betel nut chewing behaviors. Male moderate drinkers also reported more comorbidities than non-drinkers.

### Conclusions

We observed that occasional drinkers had a better socioeconomic position than non-drinkers; moderate drinkers seemed to have a worse risk profile than non-drinkers. These

---

**Data availability statement:** The data from surveys are available and can be accessed by application to the Health and Welfare Data Science Center, Ministry of Health and Welfare in Taiwan, at https://dep.mohw.gov.tw/DOS/cp-5119-59201-113.html.

**Funding:** This study was funded by the National Science and Technology Council in Taiwan. The funders had no role in study design, data collection and analysis, decision to publish, or preparation of the manuscript.

**Competing interests:** The authors declare that they have no known competing financial interests or personal relationships that could influence the work reported in this paper.

confounding patterns were different from that in Western countries and may thereby help us make causal inferences regarding alcohol and health.

## Introduction

Although excessive alcohol consumption has many negative impacts on health, moderate alcohol consumption has been linked to positive cardiovascular health in numerous studies [1–4]. Scientific communities are still uncertain about the many aspects of cause and effect between alcohol and health [5,6]. The National Institutes of Health in the United States even funded an international, long-term, randomized clinical trial (RCT) to assess the cardiometabolic effects of moderate alcohol drinking [7], given that most current evidence is based on observational studies or short-term RCTs. The Moderate Alcohol and Cardiovascular Health Trial, however, was later terminated because of internal policy concerns [8].

Confounding is a major impediment to the validity of studies that assess cause and effect between alcohol and health. There is a systematic difference among those who never drink, those who quit drinking, those who drink moderately, and those who drink heavily. For example, previous analysis of the Behavioral Risk Factor Surveillance System data assessed the prevalence of cardiovascular risk factors and confounders among non-drinkers and moderate drinkers [9]. They indicated that non-drinkers were more likely than moderate drinkers to have factors regarding demographics, social and behavioral factors, access to healthcare, and health-related factors that increased the risk of cardiovascular disease (CVD). Hence, the protective effect of moderate drinking on CVD observed in many studies may be largely ascribed to such differences.

Common approaches to dealing with confounding in epidemiological research include restriction, stratification, matching, and statistical adjustment. More advanced approaches for causal inferences include using propensity scores [10], instrumental variables [11], or regression discontinuity [12]. Another proposed causal inference approach was to compare the associations obtained from cohorts in different countries, where we know the confounding structures are different [13,14].

This approach was applied to assess the effect of breastfeeding on childhood obesity. For instance, Brion and colleagues compared the association of breastfeeding with childhood outcomes from a cohort in England with that from a cohort in Brazil [14]. Breastfeeding was less relevant to socioeconomic position in Brazil than in England. Since breastfeeding was only associated with the intelligence quotient in Brazil, the authors concluded that breastfeeding was more likely to be causally associated with the intelligence quotient. The association of breastfeeding with other outcomes (blood pressure and body mass index) in England could be due to residual confounding by socioeconomic position.

To examine the confounding structures among drinkers and non-drinkers, Au Yeung and colleagues compared the systematic differences among never, occasional and moderate drinkers in Southern China, and showed a different confounding pattern from that in Western countries [15]. Moderate drinkers had a lower socioeconomic position than never drinkers in Southern China. It is unclear whether there would be a similar confounding structure in Taiwanese people, who share common genetics and cultures with Southern Chinese. Our goal was thereby to compare the characteristics of drinkers and non-drinkers in Taiwan, using a national survey of nutrition and health.

## Methods

### Data

The Nutrition and Health Survey in Taiwan (NAHSIT) 2013-2016 was a nationwide cross-sectional survey conducted by the Health Promotion Administration, Ministry of Health and Welfare in Taiwan [16]. It aimed to assess the nutritional and health status of the Taiwanese population and provide important data for policymakers to develop health promotion strategies. That survey used a multistage, stratified, and cluster sampling design to select subjects aged 2 months and above; the sample was representative of residents in the 20 counties and cities in Taiwan. From 2013 to 2016, 11,072 study subjects participated in the face-to-face interview survey and 5,770 were aged 19 or older (up to 100). The survey collected data on a wide range of variables, such as sociodemographic factors, dietary intake, medical history, reproductive history, physical activity, smoking, and alcohol drinking. A large proportion of study subjects (86%) also underwent physical examination.

The ethics board at the Academia Sinica, Taiwan, approved the present study, and all study subjects signed the informed consent to participate. We obtained datasets for analysis from the Health and Welfare Data Science Center in Taiwan in October 1, 2021, without access to identifiable information of the study subjects.

### Independent variables

Our study focused on the following independent variables that may be related to alcohol drinking and CVD [9,15,17,18]:

▪ Sociodemographic variables: age, sex, education, marital status, and working status;

▪ Education was categorized into elementary, junior high, senior high, college, university, and graduate levels;

▪ Marital status was categorized into single, married, separated/divorced, and widowed;

▪ Working status was categorized into full time, part time, student/housewife, retired, and unemployed.

Study subjects were also asked to rate their family financial problems on a scale from 1 (not difficult at all) to 4 (very difficult).

Health behavior variables included physical activity, smoking, and betel nut chewing. The subjects' responses to the physical activity questions were transformed into the metabolic equivalent of task minutes (MET-minutes) per week; their physical activities were categorized into inactive, 1-599 MET-minutes/week, 600-1499 MET-minutes/week, and ≥ 1,500 MET-minutes/week. Smoking was categorized into non-smoker, former smoker, and smoker. Betel nut chewing was categorized into non-chewer, former chewer, and chewer. Subjects' comorbidities were assessed by self-reports regarding 28 types of health conditions (see S1 Table), and we calculated the number of comorbidities for each subject. They were also asked to rate their health status on a scale from 1 (excellent) to 5 (poor).

### Dependent variable

The dependent variable was alcohol drinking. Alcohol drinking was categorized into non-drinker, former drinker, occasional drinker, and drinker (moderate vs heavy). In the NAHSIT study, subjects were also queried about what type of drinks they had during the past month, the average frequency of alcohol drinking per week, and the quantity of alcohol drinking per occasion. Using this information—together with the alcohol by volume of each type of drink

and the specific gravity of the alcohol—we computed the amount of ethanol (g) consumed per week for the drinkers. The amount of ethenol can also be expressed as the number of standard drinks consumed per week, where one standard drink contains 10 g ethanol [15]. We further determined whether a drinker was a heavy drinker or a moderate drinker. We defined heavy drinkers to be men who consumed more than 14 drinks per week or women who consumed more than 7 drinks per week.

## Statistical analysis

Our goal was to compare the systematic difference among non-drinkers, occasional drinkers, and moderate drinkers. In our analysis, we excluded former drinkers since people may quit drinking as a result of their health conditions. To study the effects of alcohol drinking on health, most studies have not included former drinkers in order to refrain from reverse causation [17]. We excluded heavy drinkers as well because it is known that excessive alcohol drinking is harmful to health. Only moderate drinking would be considered potentially beneficial to cardiovascular health.

Our analysis was stratified by sex because the drinking pattern is so different between men and women in Asian countries [19]. The dependent variable used for regression analysis included three categories of alcohol drinking: non-drinkers, occasional drinkers, and moderate drinkers (excluding heavy drinkers). We used multinomial logistic regression to assess the associations between each independent variable and alcohol drinking, and the results were adjusted for a subject's difference in age. Since the NAHSIT study used multistage, stratified, and cluster sampling design, all our analyses accounted for proper survey weighting. Missing data on the independent variables were imputed by multiple imputation methods. We performed data analysis using Stata 15 (StataCorp LLC, College Station, TX, USA).

## Results

Table 1 shows the distribution of variables in men (n = 2,846) according to the five drinking categories; Table 2 shows the corresponding distribution in women (n = 2,833). There is a large difference in the distribution of drinking categories between men and women. Among men, 31% were non-drinkers, 7% former drinkers, 43% occasional drinkers, 13% moderate drinkers, and 5% heavy drinkers. Among women, 62% were non-drinkers, 1% former drinkers, 32% occasional drinkers, 4% moderate drinkers, and 1% heavy drinkers.

In both men and women, occasional drinkers were the youngest group. In men, the mean age (65.5 years) was much older for former drinkers than for the other four groups. For marital status among men, heavy drinkers were more likely to be separated/divorced; among women, former drinkers and heavy drinkers were more likely to be widowed.

Regarding other sociodemographic variables, occasional drinkers had the highest educational level in men and women. Occasional drinkers also reported having the fewest family financial problems; heavy drinkers reported having the most family financial problems and had the highest proportion of unemployed people.

Regarding health-related variables, all other drinkers were more likely than non-drinkers to be smokers or betel nut chewers, for both men and women. Occasional drinkers had the lowest proportion of people who rated their health poor and had the smallest mean number of comorbidities; former drinkers reported the largest mean number of comorbidities for both men and women.

After excluding former drinkers and heavy drinkers from our sample, we performed age-adjusted regression analysis. Table 3 shows the age-adjusted risk of being an occasional or moderate drinker versus a non-drinker in men, according to independent variables. A larger

**Table 1. Distribution of variables according to drinking status in men (n = 2,846).**

| Variables | Non-drinker | Former drinker | Occasional drinker | Drinker (moderate) | Drinker (heavy) |
|---|---|---|---|---|---|
| Sample size | 895 | 193 | 1232 | 379 | 147 |
| Age in years, mean±SD | 54.7 ± 20.0 | 65.5 ± 13.3 | 49.7 ± 18.0 | 53.7 ± 17.9 | 53.6 ± 15.9 |
| **Age group, %** | | | | | |
| 19-29 years | 17.0 | 2.6 | 17.1 | 12.9 | 6.8 |
| 30-39 years | 10.3 | 2.1 | 16.2 | 12.1 | 15.7 |
| 40-49 years | 9.7 | 5.7 | 15.3 | 14.8 | 20.4 |
| 50-59 years | 16.5 | 16.6 | 17.9 | 18.7 | 19.1 |
| 60-69 years | 19.0 | 30.6 | 18.2 | 20.6 | 19.7 |
| 70-79 years | 17.3 | 30.1 | 11.6 | 13.7 | 14.3 |
| ≥80 years | 10.2 | 12.4 | 3.8 | 7.1 | 4.1 |
| **Education, %** | | | | | |
| Elementary | 23.9 | 42.0 | 14.5 | 20.8 | 21.8 |
| Junior high | 13.2 | 16.1 | 10.6 | 16.1 | 19.7 |
| Senior high | 26.7 | 24.4 | 28.3 | 29.6 | 35.4 |
| College | 12.9 | 6.2 | 13.6 | 13.7 | 11.6 |
| University | 19.7 | 9.3 | 26.1 | 16.4 | 9.5 |
| Graduate | 3.7 | 1.6 | 6.9 | 3.4 | 2.0 |
| **Marital status, %** | | | | | |
| Single | 25.1 | 10.9 | 24.8 | 19.5 | 10.2 |
| Married | 64.5 | 73.6 | 66.3 | 67.3 | 75.5 |
| Separated/Divorced | 4.3 | 6.2 | 4.6 | 6.9 | 11.6 |
| Widowed | 5.9 | 9.3 | 4.3 | 6.3 | 2.0 |
| **Working status, %** | | | | | |
| Full time | 49.4 | 25.9 | 60.2 | 54.9 | 51.0 |
| Part time | 4.6 | 6.7 | 5.5 | 4.2 | 6.8 |
| Student/Housewife | 2.5 | 0 | 2.9 | 0.8 | 0.7 |
| Retired | 37.2 | 62.2 | 25.7 | 31.9 | 26.5 |
| Unemployed | 6.2 | 4.2 | 5.7 | 7.4 | 15.0 |
| **Family financial problems, %** | | | | | |
| 1 (not difficult at all) | 16.3 | 9.8 | 22.7 | 17.7 | 13.6 |
| 2 | 55.6 | 53.4 | 54.1 | 52.2 | 49.7 |
| 3 | 21.2 | 29.5 | 17.0 | 19.0 | 24.5 |
| 4 (very difficult) | 5.9 | 6.2 | 5.0 | 8.4 | 10.9 |
| **Physical activity, %** | | | | | |
| Inactive | 19.0 | 19.2 | 14.8 | 17.2 | 17.7 |
| 1-599 MET-minutes/week | 23.4 | 26.4 | 24.7 | 28.8 | 23.8 |
| 600-1499 MET-minutes/week | 26.2 | 25.4 | 25.2 | 23.8 | 21.8 |
| ≥1500 MET-minutes/week | 31.5 | 29.0 | 35.4 | 30.3 | 36.7 |
| **Smoking, %** | | | | | |
| Non-smoker | 58.8 | 16.1 | 42.1 | 24.8 | 12.2 |
| Former smoker | 22.1 | 55.4 | 25.8 | 31.9 | 22.5 |
| Smoker | 19.1 | 28.5 | 32.1 | 43.3 | 65.3 |
| **Betel nut chewing, %** | | | | | |
| Non-chewer | 86.5 | 51.3 | 77.8 | 60.7 | 44.2 |
| Former chewer | 9.1 | 39.9 | 13.2 | 25.3 | 25.9 |
| Chewer | 4.5 | 8.8 | 8.9 | 14.0 | 29.9 |

*(Continued)*

**Table 1.** (Continued)

| Variables | Non-drinker | Former drinker | Occasional drinker | Drinker (moderate) | Drinker (heavy) |
|---|---|---|---|---|---|
| Self-rated health, % | | | | | |
| 1 (excellent) | 11.0 | 11.9 | 11.1 | 11.6 | 15.0 |
| 2 | 18.2 | 15.5 | 19.3 | 18.2 | 16.3 |
| 3 | 46.3 | 39.4 | 52.6 | 50.1 | 42.2 |
| 4 | 15.8 | 18.1 | 11.1 | 12.4 | 15.0 |
| 5 (poor) | 8.0 | 14.5 | 5.2 | 6.6 | 11.6 |
| Comorbidities, mean ± SD | 1.5 ± 1.8 | 2.6 ± 1.9 | 1.3 ± 1.7 | 1.8 ± 2.0 | 1.5 ± 1.7 |

MET, Metabolic equivalent of task; SD, standard deviation.

relative risk ratio indicated that the risk of being an occasional or moderate drinker versus a non-drinker was greater for the group examined than for the reference group. For instance, the risk of being an occasional drinker versus a non-drinker was 2.42 times (95% confidence interval [CI] 1.66-3.52) higher in those who were married than those who were single. Occasional drinkers were more likely than non-drinkers to be married or widowed, have higher educational levels, fewer family financial problems, and to have smoking and betel nut chewing habits. Moderate drinkers were more likely than non-drinkers to be married, separated/divorced or widowed, have smoking and betel nut chewing habits, and more comorbidities. Retired men were less likely to be occasional or moderate drinkers.

Table 4 shows the age-adjusted risk of being an occasional or moderate drinker versus a non-drinker in women, according to independent variables. Higher educational levels and those with smoking/betel nut chewing habits were more likely to be occasional drinkers than non-drinkers. Smoking/betel nut chewing habits were also more likely to be moderate drinkers than non-drinkers. Women who worked part-time or were unemployed were less likely to be occasional or moderate drinkers.

## Discussion

The present analysis explored the systematic differences among residents in Taiwan with different drinking behaviors. There was a substantial difference in the distribution of drinking categories between men and women. Women consumed much less alcohol than men did. All in all, occasional drinkers seemed to have a higher socioeconomic position than non-drinkers and reported more smoking or betel nut chewing behaviors. Moderate drinkers reported even more smoking or betel nut chewing behaviors and also reported more comorbidities than non-drinkers.

The debate on whether moderate drinking is good for health still continues nowadays in the scientific community. A typical "J-shaped" curve is often reported to describe the relationship between alcohol drinking and risk of disease, where moderate drinkers have the lowest risk compared to non-drinkers or heavy drinkers [20]. One explanation for such a relationship is that the non-drinkers group may include some former drinkers. These former drinkers may have quit drinking because of their health problems and medication use and thus may have had a higher disease risk [21].

Our results agree with that viewpoint. We found that the former drinkers group, regardless of sex, reported the highest number of comorbidities. These study subjects were excluded from our regression analysis.

Another explanation for the lower risk of disease seen in the moderate drinkers group is that these people may have a demographic and lifestyle profile that is associated with better

**Table 2. Distribution of variables according to drinking status in women (n = 2,833).**

| Variables | Non-drinker | Former drinker | Occasional drinker | Drinker (moderate) | Drinker (heavy) |
|---|---|---|---|---|---|
| **Sample size** | 1,759 | 33 | 902 | 106 | 33 |
| **Age in year, mean ± SD** | 56.7 ± 18.1 | 56.3 ± 15.5 | 46.2 ± 16.9 | 50.8 ± 18.5 | 49.9 ± 18.3 |
| **Age group, %** | | | | | |
| 19-29 years | 10.9 | 3.0 | 22.3 | 18.9 | 18.2 |
| 30-39 years | 10.5 | 15.2 | 17.7 | 13.2 | 15.2 |
| 40-49 years | 11.4 | 12.1 | 17.1 | 16.0 | 18.2 |
| 50-59 years | 17.1 | 24.2 | 18.1 | 16.0 | 15.2 |
| 60-69 years | 22.2 | 24.2 | 13.9 | 19.8 | 15.2 |
| 70-79 years | 19.3 | 18.2 | 9.1 | 8.5 | 15.2 |
| ≥80 years | 8.6 | 3.0 | 2.0 | 7.6 | 3.0 |
| **Education, %** | | | | | |
| Elementary | 42.8 | 39.4 | 18.5 | 31.1 | 42.4 |
| Junior high | 11.7 | 27.3 | 10.4 | 17.0 | 18.2 |
| Senior high | 21.7 | 27.3 | 28.2 | 23.6 | 27.3 |
| College | 7.1 | 0 | 12.6 | 9.4 | 0 |
| University | 14.3 | 3.0 | 26.5 | 17.0 | 12.1 |
| Graduate | 2.4 | 0 | 3.8 | 1.9 | 0 |
| **Marital status, %** | | | | | |
| Single | 13.3 | 3.0 | 25.4 | 14.2 | 24.2 |
| Married | 61.6 | 60.6 | 58.2 | 62.3 | 42.4 |
| Separated/Divorced | 3.8 | 9.1 | 5.1 | 8.5 | 6.1 |
| Widowed | 21.3 | 27.3 | 11.1 | 15.1 | 27.3 |
| **Working status, %** | | | | | |
| Full time | 33.3 | 30.3 | 49.9 | 51.9 | 48.5 |
| Part time | 6.9 | 9.1 | 7.2 | 4.7 | 9.1 |
| Student/Housewife | 24.1 | 33.3 | 19.7 | 17.0 | 9.1 |
| Retired | 27.2 | 18.2 | 14.9 | 21.7 | 18.2 |
| Unemployed | 8.4 | 9.1 | 8.3 | 4.7 | 15.2 |
| **Family financial problems, %** | | | | | |
| 1 (not difficult at all) | 14.6 | 9.1 | 22.0 | 13.2 | 18.2 |
| 2 | 54.5 | 48.5 | 52.6 | 50.9 | 36.4 |
| 3 | 22.6 | 27.3 | 19.4 | 25.5 | 15.2 |
| 4 (very difficult) | 6.3 | 15.2 | 5.2 | 8.5 | 30.3 |
| **Physical activity, %** | | | | | |
| Inactive | 17.3 | 21.2 | 13.8 | 17.0 | 15.2 |
| 1-599 MET-minutes/week | 28.2 | 27.3 | 27.9 | 24.5 | 36.4 |
| 600-1499 MET-minutes/week | 26.8 | 18.2 | 27.8 | 25.5 | 24.2 |
| ≥1500 MET-minutes/week | 27.7 | 33.3 | 30.5 | 33.0 | 24.2 |
| **Smoking, %** | | | | | |
| Non-smoker | 96.1 | 54.6 | 88.3 | 77.4 | 48.5 |
| Former smoker | 1.3 | 21.2 | 3.6 | 3.8 | 9.1 |
| Smoker | 2.6 | 24.2 | 8.2 | 18.9 | 42.4 |
| **Betel nut chewing, %** | | | | | |
| Non-chewer | 98.6 | 84.9 | 95.7 | 84.9 | 69.7 |
| Former chewer | 0.8 | 6.1 | 1.3 | 3.8 | 6.1 |
| Chewer | 0.6 | 9.1 | 3.0 | 11.3 | 24.2 |

*(Continued)*

**Table 2.** (Continued)

| Variables | Non-drinker | Former drinker | Occasional drinker | Drinker (moderate) | Drinker (heavy) |
|---|---|---|---|---|---|
| Self-rated health, % | | | | | |
| 1 (excellent) | 7.2 | 6.1 | 8.0 | 13.2 | 15.2 |
| 2 | 14.5 | 15.2 | 18.7 | 12.3 | 18.2 |
| 3 | 50.2 | 42.4 | 49.2 | 45.3 | 42.4 |
| 4 | 17.2 | 24.2 | 17.1 | 15.1 | 15.2 |
| 5 (poor) | 9.6 | 12.1 | 6.5 | 14.2 | 9.1 |
| Comorbidities, mean ± SD | 1.5 ± 1.7 | 1.9 ± 2.2 | 0.9 ± 1.4 | 0.9 ± 1.2 | 1.4 ± 1.8 |

MET, Metabolic equivalent of task; SD, standard deviation.

health [22]. This relation would confound the association between moderate drinking and health.

Some studies have even reported a lower risk of cirrhosis among moderate drinkers [23] and a lower risk of developmental disorders in children of "moderate drinking" mothers [24], both suggesting a potential confounded association. In our regression analysis, we observed systematic differences among non-drinkers, occasional drinkers, and moderate drinkers, but the pattern was different. Although occasional drinkers in our sample had a better risk profile than non-drinkers, moderate drinkers seemed to have a worse risk profile.

These results were similar to what Au Yeung and colleagues found in Southern Chinese populations [15]. In their analysis, they used 26,361 participants' data in the Guangzhou Biobank Cohort Study to compare never, occasional, and moderate alcohol users. For men, they observed that moderate alcohol users had lower socioeconomic status and worse lifestyles than never users. Conversely, occasional alcohol users had higher socioeconomic status and better lifestyles than never users. Accordingly, the authors concluded that the confounding pattern for drinking and health in Southern Chinese men was different from that for men in Western countries. Studying this population would then be useful to develop causal inferences in alcohol epidemiology.

What are the implications of our study for alcohol epidemiological research? Given that moderate drinkers in Taiwan had worse risk profiles than non-drinkers, we can examine the association of drinking behaviors with cardiovascular diseases in Taiwan to draw better causal inferences. If the data still show that there is a lower risk of disease in the moderate drinkers' group, this would provide further support for the cause and effect between alcohol and cardio-vascular health.

Such an approach has been applied in various research contexts. For example, besides the study of breastfeeding and childhood outcomes by Brion et al., Murray and colleagues also used a cross-cohort comparison design to study the association between adverse pregnancy outcomes and childhood attention problems [25]. They studied the same research question in cohorts from both the United Kingdom and Brazil to learn whether they could obtain consistent results. In their case, the consistent results added weight to the causal role of fetal growth restriction on childhood attention problems.

The major strength of our study is that our data were based on a nationwide nutrition and health survey, which was representative of the noninstitutionalized residents in Taiwan. We used sophisticated sampling and weighting approaches and asked comprehensive questions to assess the exposure of alcohol drinking.

The downside is that the assessment of alcohol drinking, together with other questions regarding health conditions and behaviors, were all via self-reporting. The validity of

**Table 3.** Age-adjusted relative risk ratio for being a drinker versus non-drinker in men.

| Variables | Relative risk of being an occasional drinker vs non-drinker | Relative risk of being a drinker (moderate) vs non-drinker |
|---|---|---|
|  | Relative risk ratio (95% CI) | Relative risk ratio (95% CI) |
| **Education** |  |  |
| Elementary | Ref | Ref |
| Junior high | 1.78 (1.00 – 3.18) | 1.74 (0.90 – 3.37) |
| Senior high | 2.12 (1.35 – 3.34)* | 1.04 (0.63 – 1.72) |
| College | 2.19 (1.21 – 3.96)* | 1.07 (0.56 – 2.02) |
| University | 2.86 (1.66 – 4.92)* | 0.54 (0.28 – 1.05) |
| Graduate | 3.22 (1.60 – 6.48)* | 0.96 (0.35 – 2.62) |
| **Marital status** |  |  |
| Single | Ref | Ref |
| Married | 2.42 (1.66 – 3.52)* | 2.34 (1.27 – 4.30)* |
| Separated/Divorced | 1.65 (0.80 – 3.40) | 4.91 (1.29 – 18.61)* |
| Widowed | 2.20 (1.08 – 4.49)* | 2.84 (1.02 – 7.87)* |
| **Working status** |  |  |
| Full time | Ref | Ref |
| Part time | 0.87 (0.43 – 1.76) | 0.86 (0.32 – 2.30) |
| Student/Housewife | 0.99 (0.41 – 2.42) | 0.30 (0.06 – 1.51) |
| Retired | 0.65 (0.45 – 0.94)* | 0.53 (0.31 – 0.90)* |
| Unemployed | 0.66 (0.37 – 1.21) | 0.87 (0.42 – 1.81) |
| **Family financial problems** |  |  |
| 1 (not difficult at all) | Ref | Ref |
| 2 | 0.60 (0.43 – 0.84)* | 0.80 (0.50 – 1.28) |
| 3 | 0.52 (0.37 – 0.72)* | 1.19 (0.58 – 2.43) |
| 4 (very difficult) | 0.55 (0.26 – 1.16) | 0.96 (0.45 – 2.07) |
| **Physical activity** |  |  |
| Inactive | Ref | Ref |
| 1-599 MET-minutes/week | 1.31 (0.87 – 1.98) | 0.89 (0.52 – 1.54) |
| 600-1,499 MET-minutes/week | 1.34 (0.84 – 2.13) | 1.04 (0.56 – 1.90) |
| ≥1500 MET-minutes/week | 1.37 (0.87 – 2.17) | 0.82 (0.44 – 1.53) |
| **Smoking** |  |  |
| Non-smoker | Ref | Ref |
| Former smoker | 1.62 (1.13 – 2.32)* | 4.00 (2.26 – 7.08)* |
| Smoker | 1.94 (1.33 – 2.84)* | 7.39 (4.40 – 12.41)* |
| **Betel nut chewing** |  |  |
| Non-chewer | Ref | Ref |
| Former chewer | 2.01 (1.36 – 2.98)* | 6.02 (3.67 – 9.88)* |
| Chewer | 2.01 (1.18 – 3.44)* | 6.50 (2.57 – 16.44)* |
| **Self-rated health** |  |  |
| 1 (excellent) | Ref | Ref |
| 2 | 1.12 (0.63 – 2.01) | 1.04 (0.44 – 2.44) |
| 3 | 1.22 (0.73 – 2.02) | 1.23 (0.61 – 2.46) |
| 4 | 0.81 (0.46 – 1.43) | 1.20 (0.57 – 2.54) |
| 5 (poor) | 0.70 (0.36 – 1.38) | 0.61 (0.23 – 1.59) |
| **Comorbidities** | 1.02 (0.92 – 1.12) | 1.18 (1.05 – 1.34)* |

CI, confidence interval.

*, $p < 0.05$.

**Table 4. Age-adjusted relative risk ratio for being a drinker versus non-drinker in women.**

| | Relative risk of being an occasional drinker vs non-drinker | Relative risk of being a drinker (moderate) vs non-drinker |
|---|---|---|
| Variables | Relative risk ratio (95% CI) | Relative risk ratio (95% CI) |
| **Education** | | |
| Elementary | Ref | Ref |
| Junior high | 1.19 (0.70 – 2.02) | 1.46 (0.60 – 3.52) |
| Senior high | 1.47 (0.86 – 2.51) | 1.46 (0.70 – 3.06) |
| College | 1.82 (0.87 – 3.83) | 1.75 (0.46 – 6.64) |
| University | 2.07 (1.02 – 4.21)* | 1.43 (0.41 – 4.97) |
| Graduate | 2.65 (0.98 – 7.13) | 0.80 (0.16 – 3.93) |
| **Marital status** | | |
| Single | Ref | Ref |
| Married | 1.16 (0.77 – 1.74) | 1.65 (0.60 – 4.58) |
| Separated/Divorced | 1.47 (0.74 – 2.93) | 2.28 (0.67 – 7.75) |
| Widowed | 0.71 (0.42 – 1.19) | 1.36 (0.36 – 5.14) |
| **Working status** | | |
| Full time | Ref | Ref |
| Part time | 0.53 (0.29 – 0.95)* | 0.26 (0.07 – 0.93)* |
| Student/Housewife | 1.09 (0.73 – 1.60) | 0.85 (0.43 – 1.68) |
| Retired | 0.97 (0.59 – 1.62) | 1.00 (0.46 – 2.19) |
| Unemployed | 0.58 (0.36 – 0.94)* | 0.36 (0.05 – 2.59) |
| **Family financial problems** | | |
| 1 (not difficult at all) | Ref | Ref |
| 2 | 0.69 (0.47 – 1.03) | 0.95 (0.41 – 2.21) |
| 3 | 0.68 (0.44 – 1.04) | 1.23 (0.44 – 3.45) |
| 4 (very difficult) | 0.82 (0.41 – 1.61) | 1.33 (0.40 – 4.39) |
| **Physical activity** | | |
| Inactive | Ref | Ref |
| 1-599 MET-minutes/week | 1.01 (0.60 – 1.72) | 0.69 (0.25 – 1.86) |
| 600-1499 MET-minutes/week | 1.22 (0.74 – 2.02) | 0.67 (0.24 – 1.83) |
| ≥1500 MET-minutes/week | 1.00 (0.68 – 1.48) | 0.59 (0.29 – 1.23) |
| **Smoking** | | |
| Non-smoker | Ref | Ref |
| Former smoker | 3.48 (1.43 – 8.44)* | 3.75 (0.71 – 19.79) |
| Smoker | 3.02 (1.33 – 6.82)* | 11.82 (4.82 – 28.99)* |
| **Betel nut chewing** | | |
| Non-chewer | Ref | Ref |
| Former chewer | 1.25 (0.37 – 4.19) | 12.14 (4.60 – 32.09)* |
| Chewer | 5.60 (2.12 – 14.79)* | 16.40 (5.71 – 47.08)* |
| **Self-rated health** | | |
| 1 (excellent) | Ref | Ref |
| 2 | 0.95 (0.51 – 1.75) | 0.32 (0.10 – 1.07) |
| 3 | 0.75 (0.44 – 1.26) | 0.67 (0.24 – 1.88) |
| 4 | 0.75 (0.44 – 1.30) | 0.40 (0.12 – 1.35) |
| 5 (poor) | 0.57 (0.29 – 1.11) | 1.10 (0.31 – 3.93) |
| **Comorbidities** | 0.96 (0.89 – 1.05) | 0.81 (0.63 – 1.04) |

CI, confidence interval.

*, $p < 0.05$.

self-reporting is unlikely to be ideal and may be dependent on each subject's age, sex, and educational level. We did not have measurements for alcohol use disorders or relevant mental health conditions, but in our regression analysis we have excluded heavy drinkers.

Moreover, we made only a one-time cross-sectional assessment, but health behaviors such as alcohol drinking are dynamic processes that change over time. The type, volume, and frequency of alcohol drinking reported by our study subjects reflected only their recent use of alcohol, and this did not capture the life history of their alcohol use.

To conclude, in our sample, occasional drinkers exhibited a more favorable risk profile compared to non-drinkers, while moderate drinkers demonstrated a less favorable profile. These patterns diverge from those observed in Western populations but align closely with findings among Southern Chinese populations. To establish causal inferences on the relationship between alcohol and health, cross-country comparisons accounting for diverse confounding patterns are essential.

## Supporting information

**S1 Table. Frequency of comorbidities in men (N = 2846) and women (N = 2833).**
(DOCX)

## Author contributions

**Conceptualization:** Tsung Yu, Hsing-Yu Yang.

**Data curation:** Tzu-Jung Wong.

**Formal analysis:** Tsung Yu, Tzu-Jung Wong.

**Funding acquisition:** Hsing-Yu Yang.

**Methodology:** Tsung Yu.

**Project administration:** Tsung Yu.

**Resources:** Hsing-Yu Yang.

**Supervision:** Hsing-Yu Yang.

**Writing – original draft:** Tsung Yu.

**Writing – review & editing:** Tzu-Jung Wong, Hsing-Yu Yang.

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
