## [Decision Letter · Decision Letter 0]

15 Oct 2024

PONE-D-24-08844Characteristics of Non-Drinking and Drinking Adults in Taiwan and Their Implications in Alcohol Epidemiological StudiesPLOS ONE

Dear Dr. Yang,

Thank you for submitting your manuscript to PLOS ONE. After careful consideration, we feel that it has merit but does not fully meet PLOS ONE’s publication criteria as it currently stands. Therefore, we invite you to submit a revised version of the manuscript that addresses the points raised during the review process. The comments from the two reviewers are straightforward. 

We look forward to receiving your revised manuscript.

Kind regards,

Robert Didden

Academic Editor

PLOS ONE

Journal Requirements: When submitting your revision, we need you to address these additional requirements. 1. Please ensure that your manuscript meets PLOS ONE's style requirements, including those for file naming. The PLOS ONE style templates can be found at https://journals.plos.org/plosone/s/file?id=wjVg/PLOSOne_formatting_sample_main_body.pdf and https://journals.plos.org/plosone/s/file?id=ba62/PLOSOne_formatting_sample_title_authors_affiliations.pdf 2. Thank you for stating the following financial disclosure: "The National Science and Technology Council in Taiwan" Please state what role the funders took in the study.  If the funders had no role, please state: ""The funders had no role in study design, data collection and analysis, decision to publish, or preparation of the manuscript."" If this statement is not correct you must amend it as needed. Please include this amended Role of Funder statement in your cover letter; we will change the online submission form on your behalf. 3. In the online submission form, you indicated that "Please contact Prof. Tsung Yu for the datasets used and analyzed." All PLOS journals now require all data underlying the findings described in their manuscript to be freely available to other researchers, either 1. In a public repository, 2. Within the manuscript itself, or 3. Uploaded as supplementary information.This policy applies to all data except where public deposition would breach compliance with the protocol approved by your research ethics board. If your data cannot be made publicly available for ethical or legal reasons (e.g., public availability would compromise patient privacy), please explain your reasons on resubmission and your exemption request will be escalated for approval. 4. PLOS requires an ORCID iD for the corresponding author in Editorial Manager on papers submitted after December 6th, 2016. Please ensure that you have an ORCID iD and that it is validated in Editorial Manager. To do this, go to ‘Update my Information’ (in the upper left-hand corner of the main menu), and click on the Fetch/Validate link next to the ORCID field. This will take you to the ORCID site and allow you to create a new iD or authenticate a pre-existing iD in Editorial Manager.

Reviewers' comments:

Reviewer's Responses to Questions

**Comments to the Author**

1. Is the manuscript technically sound, and do the data support the conclusions?

Reviewer #1: Yes

Reviewer #2: Yes

2. Has the statistical analysis been performed appropriately and rigorously? 

Reviewer #1: Yes

Reviewer #2: Yes

3. Have the authors made all data underlying the findings in their manuscript fully available?

Reviewer #1: Yes

Reviewer #2: No

4. Is the manuscript presented in an intelligible fashion and written in standard English?

Reviewer #1: Yes

Reviewer #2: Yes

5. Review Comments to the Author

Reviewer #1: Title: Characteristics of Non-Drinking and Drinking Adults in Taiwan and Their Implications in Alcohol Epidemiological Studies

This study is exciting and holds promise for offering a valuable contribution to the scientific community. Following a comprehensive and critical evaluation, I wish to bring attention to several findings that require addressing. By addressing the following points or modifications, authors can enhance the clarity and impact of their research work. However, I highly recommend publishing this article once the corrections are implemented.

Comments:

Abstract:

-background or introduction is missing from the abstract; include background in the abstract.

- “Our sample included 2,846 men and 2,833 women ages 19 years and older” This line should be added to the methods, as the sample is a part of the methods.

- in conclusion, exclude “in our sample” from the line.

Introduction:

-A rigorous literature review needs to be implemented.

Methods:

-change term “Study subjects” to “Data”

- change “Data on independent variables” to “Independent variable”. Why the selected variable has been considered for this study? Whether the independent variable was selected based on previous literature or not. If yes, then cite that literature as the base of variable selection.

- “Subjects comorbidities were assessed by self-reports regarding 28 types of health conditions. They were also asked to rate their health status on a scale from 1 (excellent) to 5 (poor)”- including 28 types of disease and their percentage in a table in the manuscript. Regarding the rating of health conditions considered in the study, did the authors test the scale response using any statistical methods like Cronbach's alpha or response reliability test? -then how have comorbidities been transformed into continuous data and used as an independent variable? Fundamentally, morbidities are the outcome of certain events; for example, drinking alcohol leads to the occurrence of different morbidities. This seems to misinterpret and wrongly analyse the data; if correctly analysed, then how did comorbidities determine drinking behaviour? Carefully deal with this fallacy and just it.

- There is no indication of types of comorbidities; include it in the manuscript or in an additional file. Review the following manuscript regarding morbidities and mention it in the manuscripts- “Dolui, M., Sarkar, S., Hossain, M., & Manna, H. (2023). Demographic and socioeconomic correlates of multimorbidity due to Non-communicable diseases among adult men in India: Evidence from the nationally representative survey (NFHS-5). Clinical Epidemiology and Global Health, 23. https://doi.org/10.1016/j.cegh.2023.101376”

-change “Data on alcohol drinking” to “Dependent variable” and mention alcohol drinking as the dependent variable. In the “Data on alcohol drinking” section the line” The amount of ethanol can also be expressed as the number of standard drinks consumed per week, where one standard drink contains 10 g ethanol” need to be cited or refer the literature where it has been mentioned.

-in the Statistical Analysis section, the “Only moderate drinking would be considered potentially beneficial to cardiovascular health” needs to be modified as beneficial to indicate the positive outcome of any events. Carefully choose the words for the research article, which may create potential bias in the interpretation.

-place the lines “The dependent variable included three categories of alcohol drinking: non-drinkers, occasional drinkers, and drinkers (excluding heavy drinkers)”, in the dependent variable section.

Results:

-The sample distribution table is missing; add a sample distribution table across variables; the characteristics of the study sample must be included in the study.

-The result section needs to be modified with results values since the interpretation of Tables 2, 3, and 4 lacks results values. Interpret the results with significant values from the table only.

- the authors failed to interpret the regression table, which is a crucial part of any article, which can only determine the exposure of the outcome variable. Carefully explain the regression results.

Discussion:

- “A typical “J-shaped” curve is often reported to describe the relationship ……”, what is meant by “J-shaped” curve?

-what is the significant contribution of this study, and how will this add value to the literature?

-policy suggestions are very weak and need to be strengthened.

- the conclusion section needs to sharpen and strengthen

Reviewer #2: Thank you for the opportunity to review this manuscript. The authors present a cross-sectional analysis, comparing the demographic characteristics of drinkers and non-drinkers, using a large and representative sample in Taiwan. I have outlined several points that should be addressed before this article can be considered for publication:

1. Abstract: slightly more detail is required to explain why it is important to identify the characteristics of drinkers and non-drinkers, to address the issue of confounding factors when looking at causal inferences of alcohol and health.

2. Abstract: please state whether the data are cross sectional or longitudinal.

3. Abstract: the conclusions state that occasional drinkers had a better risk profile than non-drinkers, but this is not clear from the results section of the abstract.

4. Introduction: it may be worth referring to several studies which debunk the claim that alcohol has been linked to positive cardiovascular health, including Griswold et al (2018, The Lancet) and Stockwell et al (2024, Journal of Studies on Alcohol and Drugs)

5. Introduction: expand on what was found in the previous analysis of the Behavioural Risk Factor Surveillance System data in the same paragraph rather than starting a new paragraph.

6. Methods: please state whether the data are cross-sectional or longitudinal and include the full age range.

7. Methods: the categorisation for drinkers is confusing. The abstract refers to occasional and moderate drinkers, but the methods refers to non-drinker, former drinker, occasional drinker, and drinker (heavy drinker or not). How were occasional and moderate drinkers determined? Please clarify.

8. Methods: further justification is needed as to why former drinkers and heavy drinkers were excluded from analyses, if the aim is to compare the characteristics of drinkers and non-drinkers. This analysis would provide a more novel contribution to the literature by identifying the characteristics of never drinkers vs former drinkers, and occasional vs moderate vs heavy drinkers.

9. Results: the results section describes the characteristics of former drinkers and heavy drinkers, yet earlier it is stated that these groups were excluded from the analyses.

10. Discussion: I expected to read more limitations regarding other possible confounders that were not included in the analyses, e.g., measures of binge drinking and type of alcohol consumed, measures of mental health, whether alcohol use has changed over time.

11. Discussion: given that the study could not examine all possible confounders, and due to the cross-sectional nature of the analysis, it may be a stretch to state that these findings mean that the authors can examine the associations between drinking behaviours and cardiovascular disease to draw better causal inferences.

6. PLOS authors have the option to publish the peer review history of their article (what does this mean? ). If published, this will include your full peer review and any attached files.

**Do you want your identity to be public for this peer review?** For information about this choice, including consent withdrawal, please see our Privacy Policy .

Reviewer #1: **Yes: ** Mriganka Dolui

Reviewer #2: **Yes: ** Patricia Irizar

---

## [Author Response · Author response to Decision Letter 0]

3 Feb 2025

Dear editors,

Thank you again for the reviews of our manuscript. The comments are very helpful for us to revise and improve the manuscript. Hereby please find below point-by-point responses to the reviewers’ comments. We also attached the revised manuscript with tracked changes and a clean version.

We also have two additional statements in response to your comments:

1. Funding

This study was funded by the National Science and Technology Council in Taiwan. The funders had no role in study design, data collection and analysis, decision to publish, or preparation of the manuscript.

2. Availability of data and materials

The data from surveys are available and can be accessed by application to the Health and Welfare Data Science Center, Ministry of Health and Welfare in Taiwan.

We look forwards to seeing your further evaluation and response.

Sincerely yours,

Tsung Yu, on behalf of co-authors 

Point-by-point response to reviewer comments

1. Journal Requirements:

When submitting your revision, we need you to address these additional requirements. Please ensure that your manuscript meets PLOS ONE's style requirements, including those for file naming. The PLOS ONE style templates can be found at https://journals.plos.org/plosone/s/file?id=wjVg/PLOSOne_formatting_sample_main_body.pdf and https://journals.plos.org/plosone/s/file?id=ba62/PLOSOne_formatting_sample_title_authors_affiliations.pdf

Response:

We made the changes accordingly.

2. Thank you for stating the following financial disclosure: "The National Science and Technology Council in Taiwan". Please state what role the funders took in the study. If the funders had no role, please state: ""The funders had no role in study design, data collection and analysis, decision to publish, or preparation of the manuscript."" If this statement is not correct you must amend it as needed. Please include this amended Role of Funder statement in your cover letter; we will change the online submission form on your behalf.

Response:

We provided this statement in the cover letter.

3. In the online submission form, you indicated that "Please contact Prof. Tsung Yu for the datasets used and analyzed." All PLOS journals now require all data underlying the findings described in their manuscript to be freely available to other researchers, either 1. In a public repository, 2. Within the manuscript itself, or 3. Uploaded as supplementary information. This policy applies to all data except where public deposition would breach compliance with the protocol approved by your research ethics board. If your data cannot be made publicly available for ethical or legal reasons (e.g., public availability would compromise patient privacy), please explain your reasons on resubmission and your exemption request will be escalated for approval.

Response:

We provided a revised statement of availability of data and materials in the cover letter.

Response:

We made the changes accordingly.

5. Reviewer #1: Title: Characteristics of Non-Drinking and Drinking Adults in Taiwan and Their Implications in Alcohol Epidemiological Studies

This study is exciting and holds promise for offering a valuable contribution to the scientific community. Following a comprehensive and critical evaluation, I wish to bring attention to several findings that require addressing. By addressing the following points or modifications, authors can enhance the clarity and impact of their research work. However, I highly recommend publishing this article once the corrections are implemented.

Comments:

Abstract:

-background or introduction is missing from the abstract; include background in the abstract.

Response:

We made the changes accordingly. Please see lines 19-25 on page 2 in the manuscript with tracked changes.

6. -“Our sample included 2,846 men and 2,833 women ages 19 years and older” This line should be added to the methods, as the sample is a part of the methods.

Response:

We made the changes accordingly. Please see lines 26-28 on page 2.

7. -in conclusion, exclude “in our sample” from the line.

Response:

We made the changes accordingly. Please see line 42 on page 2.

8. Introduction:

-A rigorous literature review needs to be implemented.

Response:

We are somewhat unclear about this comment. In the Introduction section, we believe we have provided a comprehensive review of the relevant literature.

9. Methods:

-change term “Study subjects” to “Data”

Response:

We made the changes accordingly. Please see line 94 on page 5.

10. -change “Data on independent variables” to “Independent variable”. Why the selected variable has been considered for this study? Whether the independent variable was selected based on previous literature or not. If yes, then cite that literature as the base of variable selection.

Response:

We made the changes accordingly. We also provided some references. Please see lines 112-113 on page 6.

11. -“Subjects comorbidities were assessed by self-reports regarding 28 types of health conditions. They were also asked to rate their health status on a scale from 1 (excellent) to 5 (poor)”- including 28 types of disease and their percentage in a table in the manuscript. Regarding the rating of health conditions considered in the study, did the authors test the scale response using any statistical methods like Cronbach's alpha or response reliability test? -then how have comorbidities been transformed into continuous data and used as an independent variable? Fundamentally, morbidities are the outcome of certain events; for example, drinking alcohol leads to the occurrence of different morbidities. This seems to misinterpret and wrongly analyse the data; if correctly analysed, then how did comorbidities determine drinking behaviour? Carefully deal with this fallacy and justify it.

Response:

We generated a supplementary Table of comorbidities, calculating the total number of comorbidities for each individual (see lines 129-131 on page 7).

Regarding the health status scale, we are uncertain whether a reliability test or Cronbach’s alpha calculation is necessary, as only a single question was used to assess health status.

As our data are cross-sectional, reverse causality is a concern. For instance, individuals with comorbidities may avoid alcohol consumption. To address this, we excluded former drinkers from our regression analysis.

12. -There is no indication of types of comorbidities; include it in the manuscript or in an additional file. Review the following manuscript regarding morbidities and mention it in the manuscripts- “Dolui, M., Sarkar, S., Hossain, M., & Manna, H. (2023). Demographic and socioeconomic correlates of multimorbidity due to Non-communicable diseases among adult men in India: Evidence from the nationally representative survey (NFHS-5). Clinical Epidemiology and Global Health, 23. https://doi.org/10.1016/j.cegh.2023.101376”

Response:

We added this reference in the manuscript accordingly (reference number 18).

13. -change “Data on alcohol drinking” to “Dependent variable” and mention alcohol drinking as the dependent variable. In the “Data on alcohol drinking” section the line” The amount of ethanol can also be expressed as the number of standard drinks consumed per week, where one standard drink contains 10 g ethanol” need to be cited or refer the literature where it has been mentioned.

Response:

We made the changes (see lines 133-134 on page 7) and refer to a reference accordingly (reference number 15).

14. -in the Statistical Analysis section, the “Only moderate drinking would be considered potentially beneficial to cardiovascular health” needs to be modified as beneficial to indicate the positive outcome of any events. Carefully choose the words for the research article, which may create potential bias in the interpretation.

Response:

We are unsure about the specific concern raised in this comment. Our statement, “Only moderate drinking would be considered potentially beneficial to cardiovascular health,” is based on existing research that suggests moderate alcohol consumption may have cardiovascular benefits. Please refer to references number 1–4 for supporting evidence.

15. -place the lines “The dependent variable included three categories of alcohol drinking: non-drinkers, occasional drinkers, and drinkers (excluding heavy drinkers)”, in the dependent variable section.

Response:

This statement only applies to the statistical analysis section and we have clarified it (see lines 156-158 on page 8).

16. Results:

-The sample distribution table is missing; add a sample distribution table across variables; the characteristics of the study sample must be included in the study.

Response:

We do not completely understand this comment. Tables 1 and 2 are the characteristics of the study sample, stratified by sex.

17. -The result section needs to be modified with results values since the interpretation of Tables 2, 3, and 4 lacks results values. Interpret the results with significant values from the table only.

Response:

Because readers can find values in the Tables, we did not mention these values in the text. We added some sentences to explain the coefficients in the regression models. Please see lines 192-196 on page 9.

18. -the authors failed to interpret the regression table, which is a crucial part of any article, which can only determine the exposure of the outcome variable. Carefully explain the regression results.

Response:

Please see our response to point 17.

19. Discussion:

- “A typical “J-shaped” curve is often reported to describe the relationship ……”, what is meant by “J-shaped” curve?

Response:

Please refer to reference number 20. Below is a figure in the paper.

20. -what is the significant contribution of this study, and how will this add value to the literature?

Response:

The implication of our study is more methodological, adding value to the causal inference approaches. Please find our discussion in lines 251-256 on page 12.

21. -policy suggestions are very weak and need to be strengthened.

Response:

Please find our response to point 20.

22. the conclusion section needs to sharpen and strengthen

Response:

We made the changes accordingly. Please see lines 280-285 on page 13.

23. Reviewer #2: Thank you for the opportunity to review this manuscript. The authors present a cross-sectional analysis, comparing the demographic characteristics of drinkers and non-drinkers, using a large and representative sample in Taiwan. I have outlined several points that should be addressed before this article can be considered for publication:

Abstract: slightly more detail is required to explain why it is important to identify the characteristics of drinkers and non-drinkers, to address the issue of confounding factors when looking at causal inferences of alcohol and health.

Response:

Please find our response to point 5.

24. Abstract: please state whether the data are cross sectional or longitudinal.

Response:

We made the changes accordingly. Please see line 26 on page 2.

25. Abstract: the conclusions state that occasional drinkers had a better risk profile than non-drinkers, but this is not clear from the results section of the abstract.

Response:

We made revisions to the conclusion. Please find lines 42-44 on pages 2-3.

26. Introduction: it may be worth referring to several studies which debunk the claim that alcohol has been linked to positive cardiovascular health, including Griswold et al (2018, The Lancet) and Stockwell et al (2024, Journal of Studies on Alcohol and Drugs)

Response:

We added these citations accordingly. Please find citations 5 and 6.

27. Introduction: expand on what was found in the previous analysis of the Behavioural Risk Factor Surveillance System data in the same paragraph rather than starting a new paragraph.

Response:

We made the changes accordingly. Please find line 64 on page 4.

28. Methods: please state whether the data are cross-sectional or longitudinal and include the full age range.

Response:

We made the changes accordingly. Please find line 96 on page 5 and line 103 on page 6.

29. Methods: the categorisation for drinkers is confusing. The abstract refers to occasional and moderate drinkers, but the methods refers to non-drinker, former drinker, occasional drinker, and drinker (heavy drinker or not). How were occasional and moderate drinkers determined? Please clarify.

Response:

In our survey, we categorized the participants into five drinking categories, non-drinker, former drinker, occasional drinker, moderate drinker and heavy drinker. In our regression analysis, we only included non-drinker, occasional drinker and moderate drinker. We clarified this in the Methods section. Please see lines 133-154 on pages 7-8.

30. Methods: further justification is needed as to why former drinkers and heavy drinkers were excluded from analyses, if the aim is to compare the characteristics of drinkers and non-drinkers. This analysis would provide a more novel contribution to the literature by identifying the characteristics of never drinkers vs former drinkers, and occasional vs moderate vs heavy drinkers.

Response:

Please find our explanation in lines 148-154 on page 8.

31. Results: the results section describes the characteristics of former drinkers and heavy drinkers, yet earlier it is stated that these groups were excluded from the analyses.

Response:

Please find our response to point 29. We excluded former drinkers and heavy drinkers only in regression analysis.

32. Discussion: I expected to read more limitations regarding other possible confounders that were not included in the analyses, e.g., measures of binge drinking and type of alcohol consumed, measures of mental health, whether alcohol use has changed over time.

Response:

We thank the reviewer for this suggestion. Please see lines 269-279 on pages 12-13.

33. Discussion: given that the study could not examine all possible confounders, and due to the cross-sectional nature of the analysis, it may be a stretch to state that these findings mean that the authors can examine the associations between drinking behaviours and cardiovascular disease to draw better causal inferences.

Response:

Please see our revised conclusion (lines 280-285 on page 13). Our point was that, to establish causal inferences on the relationship between alcohol and health, cross-country comparisons accounting for diverse confounding patterns are essential.

---

## [Decision Letter · Decision Letter 1]

10 Feb 2025

PONE-D-24-08844R1Characteristics of Non-Drinking and Drinking Adults in Taiwan and Their Implications in Alcohol Epidemiological StudiesPLOS ONE

Dear Dr. Yang,

Thank you for submitting your manuscript to PLOS ONE. I would be happy to accept your paper provided you address some final minor comments from one reviewer.

We look forward to receiving your revised manuscript.

Kind regards,

Robert Didden

Academic Editor

PLOS ONE

Journal Requirements:

Reviewers' comments:

Reviewer's Responses to Questions

**Comments to the Author**

1. If the authors have adequately addressed your comments raised in a previous round of review and you feel that this manuscript is now acceptable for publication, you may indicate that here to bypass the “Comments to the Author” section, enter your conflict of interest statement in the “Confidential to Editor” section, and submit your "Accept" recommendation.

Reviewer #1: All comments have been addressed

Reviewer #2: (No Response)

2. Is the manuscript technically sound, and do the data support the conclusions?

Reviewer #1: Yes

Reviewer #2: Yes

3. Has the statistical analysis been performed appropriately and rigorously? 

Reviewer #1: Yes

Reviewer #2: Yes

4. Have the authors made all data underlying the findings in their manuscript fully available?

Reviewer #1: Yes

Reviewer #2: Yes

5. Is the manuscript presented in an intelligible fashion and written in standard English?

Reviewer #1: Yes

Reviewer #2: Yes

6. Review Comments to the Author

Reviewer #1: Thank you very much for allowing me to review this research article. The authors have addressed my comments and have incorporated them into the revised manuscript. I appreciate the time and effort the authors have towards manuscript revision. I highly recommended to accept this manuscript for publication.

Reviewer #2: I thank the authors for taking the time to revise their manuscript to address the reviewer concerns. The authors have sufficiently addressed most of my concerns. However, I am still unsure on the rationale for excluding former drinkers and heavy drinkers from the main analyses. I believe this paper would be strengthened substantially (particularly as the authors even discuss the issue with former drinkers sometimes being grouped with non-drinkers) by including former drinkers as a separate category in the multinomial analyses.

In the abstract (and results), the percentages of each category of drinkers for men do not equal 100%.

The analysis section needs further clarification. Some analyses were conducted with the full sample, comparing descriptive statistics. Then, former drinkers and heavy drinkers were excluded for the main analyses. However, given the rationale of aiming to understand confounders of the relationship between alcohol and health, and given that former drinkers are sometimes included with non-drinkers and heavy drinkers are sometimes included with moderate drinkers, I think this analysis would be strengthened by including former drinkers and heavy drinkers in the analysis.

When talking about the results, it is confusing when the authors say "occasional (or moderate)" as these are distinct categories.

By excluding former drinkers from the analysis, the discussion relating to former drinkers often being grouped with non-drinkers is limited. This analysis has the potential to add to the evidence base by clarifying differences between non-drinkers and former-drinkers.

7. PLOS authors have the option to publish the peer review history of their article (what does this mean? ). If published, this will include your full peer review and any attached files.

**Do you want your identity to be public for this peer review?** For information about this choice, including consent withdrawal, please see our Privacy Policy .

Reviewer #1: **Yes: ** Mriganka Dolui

Reviewer #2: **Yes: ** Patricia Irizar

---

## [Author Response · Author response to Decision Letter 1]

10 Feb 2025

Dear editors,

Thank you again for the reviews of our manuscript. The comments are very helpful for us to revise and improve the manuscript. Hereby please find below point-by-point responses to the reviewers’ comments. We also attached the revised manuscript with tracked changes and a clean version.

We look forwards to seeing your further evaluation and response.

Sincerely yours,

Tsung Yu, on behalf of co-authors 

Point-by-point response to reviewer comments

1. Reviewer #1: Thank you very much for allowing me to review this research article. The authors have addressed my comments and have incorporated them into the revised manuscript. I appreciate the time and effort the authors have towards manuscript revision. I highly recommended to accept this manuscript for publication.

Response:

We thank the reviewer for these comments.

2. Reviewer #2: I thank the authors for taking the time to revise their manuscript to address the reviewer concerns. The authors have sufficiently addressed most of my concerns. However, I am still unsure on the rationale for excluding former drinkers and heavy drinkers from the main analyses. I believe this paper would be strengthened substantially (particularly as the authors even discuss the issue with former drinkers sometimes being grouped with non-drinkers) by including former drinkers as a separate category in the multinomial analyses.

Response:

We appreciate the reviewer’s insightful comments. Our objective was to examine confounding patterns when comparing moderate drinking—often associated with better cardiovascular health in many studies—with non-drinking. In this type of analysis, it is more appropriate to exclude former and heavy drinkers. Former drinkers may have abstained due to health issues (a reverse causality problem), while heavy drinking is universally discouraged and commonly linked to poorer health outcomes. A similar study also implemented these exclusions (Au Yeung, SL, et al. “Systematic differences among never, occasional and moderate alcohol users in southern China, and its use in alcohol research: a cross-sectional study.” J Epidemiol Community Health 67.12 (2013): 1054-1060.).

“Heavy alcohol users (drinking more than moderate amounts) were excluded from the analysis as the effect of heavy alcohol use on health is less controversial and is seen in non-Western settings. Moreover, the association of heavy drinking with ill health is less likely to be confounded by higher socioeconomic position and healthy lifestyles than the association of moderate drinking with good health. Former users were also excluded because the relationship of former users and health depends on their previous alcohol use status, and thus, differences in the relationship between former use and health across settings could be a reflection of different composition of former alcohol users. In this study, former users are likely to be former occasional users.”

3. In the abstract (and results), the percentages of each category of drinkers for men do not equal 100%.

Response:

Thank you for pointing this out. This is a results of rounding errors.

4. The analysis section needs further clarification. Some analyses were conducted with the full sample, comparing descriptive statistics. Then, former drinkers and heavy drinkers were excluded for the main analyses. However, given the rationale of aiming to understand confounders of the relationship between alcohol and health, and given that former drinkers are sometimes included with non-drinkers and heavy drinkers are sometimes included with moderate drinkers, I think this analysis would be strengthened by including former drinkers and heavy drinkers in the analysis.

Response:

Please see our response to point 2.

5. When talking about the results, it is confusing when the authors say "occasional (or moderate)" as these are distinct categories.

Response:

We have revised the wording to “occasional or moderate drinker.”

6. By excluding former drinkers from the analysis, the discussion relating to former drinkers often being grouped with non-drinkers is limited. This analysis has the potential to add to the evidence base by clarifying differences between non-drinkers and former-drinkers.

Response:

Please see our response to point 2.

---

## [Editor Report · Decision Letter 2]

19 Feb 2025

Characteristics of Non-Drinking and Drinking Adults in Taiwan and Their Implications in Alcohol Epidemiological Studies

PONE-D-24-08844R2

Dear Dr. Yang,

We’re pleased to inform you that your manuscript has been judged scientifically suitable for publication and will be formally accepted for publication once it meets all outstanding technical requirements.

Kind regards,

Robert Didden

Academic Editor

PLOS ONE
---

## [Editor Report · Acceptance letter]

PONE-D-24-08844R2

PLOS ONE

Dear Dr. Yang,

I'm pleased to inform you that your manuscript has been deemed suitable for publication in PLOS ONE. Congratulations! Your manuscript is now being handed over to our production team.

Kind regards,

on behalf of

Professor Robert Didden

%CORR_ED_EDITOR_ROLE%

PLOS ONE